# Equivariant Networks for Crystal Structures

**Sékou-Oumar Kaba, Siamak Ravanbakhsh**
School of Computer Science,
McGill University
Mila - Quebec Artificial Intelligence Institute
{kabaseko@mila.quebec, siamak@cs.mcgill.ca}

## Abstract

Supervised learning with deep models has tremendous potential for applications in materials science. Recently, graph neural networks have been used in this context, drawing direct inspiration from models for molecules. However, materials are typically much more structured than molecules, which is a feature that these models do not leverage. In this work, we introduce a class of models that are equivariant with respect to crystalline symmetry groups. We do this by defining a generalization of the message passing operations that can be used with more general permutation groups, or that can alternatively be seen as defining an expressive convolution operation on the crystal graph. Empirically, these models achieve competitive results with state-of-the-art on property prediction tasks.

## 1 Introduction

Deep learning has seen remarkable applications in computational chemistry, both on the side of molecular property prediction and molecule generation. For small molecules, these methods are close to a level of precision that would make them suitable for practical applications [46]. However, less attention has been put to the problem of designing neural network architectures for the larger assemblies of atoms that constitute materials. This is a crucial problem, as deep learning could advantageously complement computationally demanding ab initio simulation methods like DFT [38, 28].

In this paper, we address the problem of designing equivariant layers and use them for supervised learning on materials. We focus on crystals, materials characterized by the ordered arrangement of their atoms in lattices. Crystalline materials are largely present around us and essential in technological applications, as they include a large number of metals, ceramics, and salts, among others [3]. They can be described as the regular repetition of a set of atoms, called the *unit cell*, in all directions of space. These patterns are analogous to wallpaper patterns, for which the repeated structure is an image. Crystals are characterized by a high degree of symmetry which is fundamental in understanding their physical properties. Consequently, we suggest that equivariant deep learning provides an appropriate framework to design function approximators for crystals.

Graph Neural Networks (GNNs) have recently been proposed for supervised prediction tasks on crystals [54, 70, 12]. We hypothesize here that these models may fall short in exploiting crystal symmetry and that structural information is lost when mapping a crystal structure to a graph. In particular, GNNs are equivariant to the permutation of atoms given as input to the model. This condition is overly restrictive and amounts to forgetting about the *ordered* nature of a crystal. We propose to use models equivariant to a product of groups $G_\Lambda \times S_C$, where $G_\Lambda$ acts at the level of the *Bravais lattice*, the underlying periodic grid of the crystal, and $S_C$ on the unit cells. We show that our proposed equivariant architecture is more expressive than GNNs on this data structure.

36th Conference on Neural Information Processing Systems (NeurIPS 2022).

By using the crystal structure, our approach amounts to defining a group-equivariant convolution kernel on the crystal in a way that is completely analogous to convolutional neural networks (CNNs). This convolution is defined on a graph associated with a crystal structure. Our contributions are the following: 1) We derive different equivariant models based on the group-theoretical properties of crystals. 2) We show some of the limitations of GNNs for crystal data and propose an alternative data structure to be used with our architecture. 3) We perform a rigorous analysis, cleaning, and processing of the Materials Project database [33] and share the resulting processed dataset to serve as a benchmark for materials applications. 4) We perform experimental tests of our models on the Materials Project database and report results comparable to or better than baselines.

## 2 Related works

**Equivariant neural networks** It is well known that neural networks have to incorporate inductive biases to be useful in practice [67, 7]. Using models that are invariant or equivariant to the symmetry of the data has proven to be a particularly important inductive bias to promote generalization [8]. The first notable application of this idea was the CNN, for which each convolution layer is equivariant to translation [42]. An alternative parameter-sharing view also appears in early works [57]; while more recent equivariant networks have used both convolutional [13, 40, 15, 19], and parameter-sharing view [51, 24]. A few notable symmetries considered in recent years are rotation in image and volumetric data [13, 68, 65], permutation symmetry in sets [72, 50] and graphs [39, 44], as well as Euclidean [60, 10], and rotational symmetry [14, 2, 21, 55, 23]. In this work, we build on foundational work on hierarchical symmetries [45, 64]

**Deep learning for materials** The increasing availability of large materials datasets from high-throughput calculations [18, 33, 36, 47, 31, 11], makes deep learning more and more relevant for materials science. Following the successes of GNNs on molecular data, similar models have been proposed for materials as alternatives to methods based on feature engineering. Note that in what follows, we refer to GNNs in a general sense that includes message passing neural networks (MPNNs) [26].Many variants of GNNs exist [35, 62, 71, 44], the underlying idea being for each node to aggregate features of neighboring nodes in a permutation invariant, or in the case of [39] in an equivariant way. The CGCNN [70] and MEGNet models [12] rely on mapping crystal structures to graphs and applying GNNs to obtain a prediction. For the SchNet model, the correspondence with graphs is less explicit [54], but still present. Other approaches combine permutation equivariance with E(3)-equivariance to design models for molecules and materials [5, 53].

## 3 Background on crystal symmetry

We first start by introducing some principles of crystallography that will be used to derive our main results. A more comprehensive treatment can also be found in the references [20, 59], whereas basics of group theory are covered in [52] for example.

**Lattices** A crystal can be described as the periodic and infinite repetition of a pattern in all directions of space. Crystals are conveniently described using lattices as their underlying structures.

An $n$-dimensional *lattice* $\Lambda$ can be defined as the set of integral combinations of the linearly independent *lattice basis vectors* $\mathbf{a}_i \in \mathbb{R}^n$:

$$\Lambda \doteq \left\{ \sum_i^n m_i \mathbf{a}_i \mid m_i \in \mathbb{Z} \right\}. \tag{1}$$

The lattice is entirely specified by its basis vectors $\mathbf{a}_i$. A lattice is associated with a group of translations $T_\Lambda$ for which the multiplication rule is addition. This captures the translational symmetry of a crystal. A lattice $\Lambda$ also defines subsets of $\mathbb{R}^n$ called *unit cells*. These subsets have the property of tilling the space when translated by lattice vectors. Of particular importance is the primitive cell $U$ for the basis $\mathbf{a}_i$:

$$U \doteq \left\{ \sum_i^n x_i \mathbf{a}_i \mid 0 \leq x_i < 1 \right\}. \tag{2}$$

In a material, the unit cell comprises a set of atomic positions $S = \{(Z_i, \mathbf{x}_i) \mid \mathbf{x}_i \in U\}$, where the integer $Z_i$ is the atomic number and $\mathbf{x}_i$ the position of the atom. $S$ can contain an arbitrary number of atoms and must not possess a particular structure. Together with the lattice $\Lambda$, the atomic positions provides a complete description of the crystal structure.

It is often useful to define the concept of a *sublattice*. A sublattice $\Lambda_P$ of $\Lambda$ is a lattice with basis vectors $\mathbf{b}_i$ such that $\Lambda_P \subset \Lambda$. Correspondingly, the *supercell* $C_P$ is the unit cell associated with the sublattice $\Lambda_P$, for which $C_P \supset U$. The full lattice is generated by translations of the sublattice by a set of *centring vectors* $\{\mathbf{0} \ldots \mathbf{v}_s\}$. More formally, the sublattice is associated with a normal subgroup $T_{\Lambda_P}$ of lattice translations, which specifies a coset decomposition of the original lattice $T_\Lambda = (\mathbf{0} + T_{\Lambda_P}) \cup \cdots \cup (\mathbf{v}_s + T_{\Lambda_P})$. The centring vectors are coset representatives with respect to that decomposition. It is clear that the centering vectors are also the set of lattice points contained in the supercell $C_P$.

---

**Example 3.1 (Graphene)** In graphene, carbon atoms are arranged in a two-dimensional crystal with honeycomb structure. The underlying lattice $\Lambda$ has basis vectors $\mathbf{a}_1 = \frac{a}{2}\begin{bmatrix} 3 & \sqrt{3} \end{bmatrix}$ and $\mathbf{a}_2 = \frac{a}{2}\begin{bmatrix} 3 & -\sqrt{3} \end{bmatrix}$, where $a$ is a lattice constant. The set of atoms in the unit cell is $S = \left\{ \left(6, -\frac{a}{2}\begin{bmatrix} 1 & 0 \end{bmatrix}\right), \left(6, \frac{a}{2}\begin{bmatrix} 1 & 0 \end{bmatrix}\right) \right\}$.

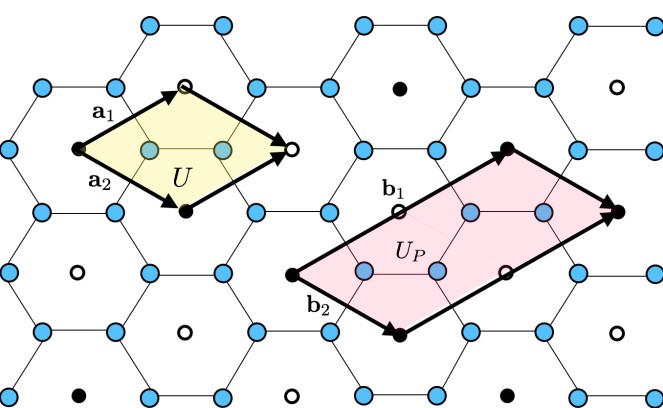

Figure 1: Graphene crystal structure

In Fig. 1, blue dots represent carbon atoms. Black and white dots identify points of lattice $\Lambda$ with unit cell $U$. Black points identify a sublattice $\Lambda_P$ defined by basis vectors $\mathbf{b}_1$ and $\mathbf{b}_2$ and with supercell $U_P$. The centring vectors $\{\mathbf{0}, \mathbf{a}_1\}$ correspond to lattice points contained in the supercell.

---

**Space groups** Symmetry plays a major role in the description of crystals. This symmetry is often directly visible through facets in naturally occurring crystals. Mathematically, it is described by a *space group* $G$, the set of isometries that maps a crystal structure to itself. As isometries, space groups are subgroups of the Euclidean group $E(n)$. A space group element can be described as a tuple $(\mathbf{W}, \mathbf{t})$, where $\mathbf{W}$ is the linear part of the transformation and $\mathbf{t}$ a translation. An element maps a vector $\mathbf{x} \in \mathbb{R}^n$ to $\mathbf{W}\mathbf{x} + \mathbf{t}$. The multiplication of space group elements is therefore given by $(\mathbf{W}_1, \mathbf{t}_1)(\mathbf{W}_2, \mathbf{t}_2) = (\mathbf{W}_1\mathbf{W}_2, \mathbf{W}_2\mathbf{t}_1 + \mathbf{t}_2)$. The point group $P$ of a space group is the group obtained from linear part operations in $G$, which will in general be rotations and reflections. Considering only elements in $G$ for which the linear part is identity $(\mathbf{I}, \mathbf{t})$, we obtain the translation subgroup of the space group $T$. It is a normal subgroup, which allows defining the factor group $G/T$ isomorphic to the point group $P$.

The crystallographic restriction theorem guarantees that only certain finite groups are valid point groups of space groups [17]. In particular, in 2 and 3 dimensions, only $n$-fold rotations with $n \in \{2, 3, 4, 6\}$ are allowed. Two space groups $G$ and $G'$ are said to belong to be of same type if they can be related by a change of coordinate system. There are 17 space group types in 2 dimensions and 230 in 3 dimensions [59]. For a lattice $\Lambda$, we call the *Bravais group* of the lattice $P_\Lambda$ the set of linear isometries that map $\Lambda$ to itself. Bravais groups provide a way to classify lattices according to their

symmetry. We say that two lattices $\Lambda$ and $\Lambda'$ belong to the same *Bravais type* if their Bravais groups are the same matrix groups when written for suitable basis vectors of each lattice. The 5 Bravais types in 2 dimensions and the 14 in 3 dimensions are enumerated in the Appendix. The full symmetry group $G_\Lambda$ of lattice a $\Lambda$ is given by the semidirect product of its translation group with its Bravais group : $G_\Lambda = T_\Lambda \rtimes P_\Lambda$.

Consider the space group $G$ a crystal structure with lattice $\Lambda$ and unit cell $U$. From the definition of a crystal structure, it is clear that the translation subgroup of $G$ will be $T_\Lambda$. However, the point group of $G$ will, in general, not be $P = P_\Lambda$. This is because the unit cell may have less symmetry than the underlying lattice, which will result in $P \subseteq P_\Lambda$ (see example 3.2). This is the reason why the number of space groups is much larger than the number of Bravais lattices.

---

**Example 3.2 (Wallpaper pattern)** The wallpaper in Figure 2 is described by a square Bravais lattice $\Lambda$ and unit cell $U$ with 4-fold rotational symmetry. The square lattice has a symmetry group $P4m$, with $D_4$ Bravais group. However, the unit cell reduces the symmetry of the overall pattern since it does not have reflection axes. The symmetry group of the wallpaper is, therefore, $P4$, with point group $C_4$.

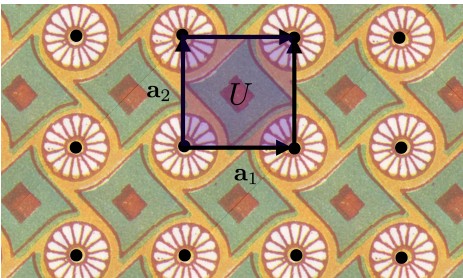

Figure 2: Egyptian wallpaper [34]

---

# 4    Limitations of GNNs

In this section, we examine some of the limitations of GNNs on crystalline data, motivating our model by addressing these limitations. Typical models for crystalline data [70, 12] will build a graph from atoms in the unit cell and assign a feature vector $\mathbf{h}_i \in \mathbb{R}^n$ to each. Edge features encoding distance might also be used. For convenience, this graph can be represented as a sparse tensor $\mathbf{H} \in \mathbb{R}^{N^2 \times n}$, where $N$ is the number of atoms considered, and in which node and edge features are encoded on diagonal and off-diagonal entries, respectively. A GNN is a function approximator $f : \mathbb{R}^{N^2 \times n} \to \mathbb{Y}$ that produce a prediction, by successive application of layers $\phi \colon \mathbb{R}^{N^2 \times n_\ell} \to \mathbb{R}^{N^2 \times n_{\ell+1}}$, where $\ell$ is the layer index.

**Expressivity**    GNNs are usually stack of layers that are permutation equivariant

$$\phi\left((\mathbf{P}(g) \otimes \mathbf{P}(g))\mathbf{H}\right) = (\mathbf{P}(g) \otimes \mathbf{P}(g))\phi\left(\mathbf{H}\right), \forall g \in S_N, \mathbf{H} \in \mathbb{R}^{N^2 \times n}, \tag{3}$$

where $\mathbf{P}(g)$ is the permutation matrix associated with the group element $g$. While $\mathbf{P}(g)$ permutes nodes, $\mathbf{P}(g) \otimes \mathbf{P}(g)$ permutes the vectorized adjacency matrix. Our use of Kronecker product is due to the equality $\mathrm{vec}(\mathbf{P}\mathbf{A}\mathbf{P}^\top) = (\mathbf{P} \otimes \mathbf{P})\mathrm{vec}(\mathbf{A})$ for $\mathbf{A} \in \mathbb{R}^{N \times N}$.

This captures the fact that these models are designed to be insensitive to node ordering in $\mathbf{H}$. The material is, in a sense, treated as if it was a molecule. Much of the crystal structure is forgotten because it cannot be captured by ordering the nodes in a specific way and is only encoded in positions or distances. We argue that permutation equivariance is an overly strong requirement and was mainly used for practical reasons. When possible, it should be more beneficial to use a model equivariant to the actual symmetry of the data; in this case, the crystal space groups, which will, in general, be much smaller than the symmetric group. Being equivariant to a smaller group results in less restrictive parameter sharing and a more expressive model.

Note that this requirement is different from that of $E(3)$-equivariance characteristic of some architectures [60, 53, 5]. In these models, the $E(n)$ group only acts on the position $\mathbf{x}_i \in \mathbb{R}^3$ part of each

atom's feature vector. By contrast, the symmetry group of a crystal structure maps the crystal to itself, and its action is a permutation. It therefore offers an alternative to building more powerful architectures for these systems that does not suppose that coordinate information is available. In particular, this approach is more suitable for abstract condensed matter systems, like spin and free-fermion lattice models, in which coordinates are not relevant. Moreover, space groups are subgroups of the Euclidean group; equivariance to space groups is thus less restrictive. In this work, we choose to concentrate on space group equivariance, but we still provide an $E(n)$-equivariant version of our architecture, which is a straightforward extension of [53] in Appendix A.5. Note that equivariance (and not only invariance) is important, as expressive invariant functions can be built by composing equivariant layers with an output pooling layer. Equivariant functions can also be used to predict local properties like magnetization and charge distribution for example.

**Invariance and symmetry breaking**   Even if the arrangement of atoms in a crystal is symmetric, this does not have to carry over to all local properties of the material. *Spontaneous symmetry breaking* is common in materials and crucial to describing phenomena such as magnetism and superconductivity [41, 6]. Building a graph only at the unit cell level does not allow to capture local properties that differ across unit cells. Using a supercell of multiple unit cells does not suffice to solve this problem since permutation equivariant models have the property that equal input elements will be mapped to equal outputs elements [58, 73].

# 5   Input representation

**Supercell**   Following the arguments of Section 4 we consider the set of atoms in a supercell instead of only in the unit cell, and add explicit symmetry breaking to increase the representational power of the model. Since this increases the computational complexity of the method, we choose to keep the supercells small and define them with sublattice vectors $\mathbf{b}_i = 2\mathbf{a}_i$. The supercell is therefore 8 times larger than the unit cell, with centring vectors $\left\{ \sum_i^3 m_i \mathbf{a}_i \mid m_i \in \{0, 1\} \right\}$. For each atom, we build a feature vector with a one-hot encoding of the atomic number and a one-hot encoding identification of the unit cell it belongs to using the corresponding centering vector. We keep track of the index of each atom within the unit cell, and the index of the atoms' unit cell $\mathbf{h}_{(a,i)} = [\text{onehot}\,(Z_i)\,, \text{onehot}\,(a)]$, where $a \in \{1, \ldots, 8\}$ and $i \in C$, where $C$ is the number of atoms in the unit cell. The encoding of the unit cell allows to break the symmetry between atoms mapped into each other by lattice translations.

**Graph**   We construct a graph from atoms in the supercell, encoding relative distances between atoms as edge features. Inspired by [54], an edge feature vector is built from the distance between atoms $d_{ij} = \|\mathbf{x}_i - \mathbf{x}_j\|$ as $\mathbf{e}_{ij} = \exp\left(-\gamma \left(d_{ij} - \boldsymbol{\mu}\right)^2\right)$. The vector of Gaussian centers $\boldsymbol{\mu}$ and $\gamma$ are hyperparameters. We use this approach to facilitate comparaison to previous works [54, 70], although Bessel encondings [37] could also be considered. It can be seen as "soft" binning of interatomic distances. Using this approach, we use only distance features in contrast to methods that use position vectors. This has the benefit of simplicity while still allowing a complete description of the input structure [66, 4, 63].

Sparsity has proven to be a useful inductive bias in graph representation learning [25], and it is also beneficial in reducing computational complexity. However, atomic bonds are not unambiguously defined in crystals [16, 1]. We choose to follow a similar approach to [32]: an edge is drawn between atoms if they share a Voronoi face and if the distance between atoms is smaller than the sum of atomic Cordero radii plus a cutoff $\Delta = 0.5\text{Å}$. This approach has the advantage of being physically sound and producing graphs that are relatively sparse. We provide more details on the graph-building strategy and compute metrics on the resulting graphs in Appendix A.1.

To preserve translational invariance for atoms at the boundary of the supercell, edges are initially also drawn to atoms outside the supercell. Then, if an edge points outside the supercell, its head is mapped to the corresponding representative node inside the supercell. This is analogous to circular padding in image processing.

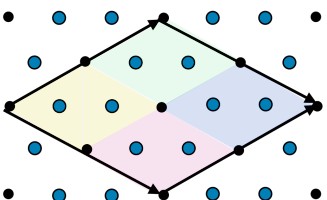

(a) Identification of the supercell. The sublattice vectors are $\mathbf{b}_i = 2\mathbf{a}_i$
.

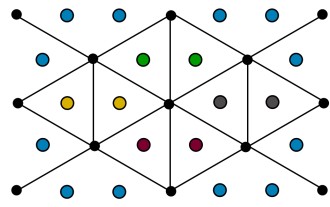

(b) Voronoï tessellation.

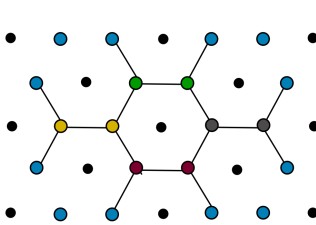

(c) Drawing edges between atoms that share Voronoï faces. The additional condition related to the Cordero radius is not relevant here.

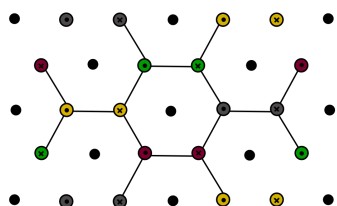

(d) Periodization of the graph. Edges that point to nodes outside the supercell are mapped back to the corresponding atom inside the supercell. Markings show identical atoms.

Figure 3: Building a graph for the graphene crystal structure from Example 3.1.

## 6 Equivariant crystal networks

**Product groups** Consider a crystal structure with space group $G$ and in which each atom has a feature vector $\mathbf{h}_i$. Then the space group acts as a permutation of the input atoms

$$g\mathbf{h}_{(a,i)} = \mathbf{h}_{\left(\pi_g^S(a), \pi_g^U(i)\right)} \forall g \in G, \tag{4}$$

where $\pi_g^S$ and $\pi_g^U$ are the permutations associated with group element $g$ on the supercells and within the unit cells respectively.

If a dataset contains only samples that share the same crystal structure, then a model equivariant with respect to $G$ can readily be used. However, the case of a dataset with multiple different crystal structures that would be used in a typical supervised learning setting is more challenging for two reasons. First, samples may have different space groups, which would require different models, each be trained with a fraction of the data. Second, the group action may be different even for the same space group. This is because the unit cells may have different structures and numbers of atoms.

A solution to address these issues is to consider equivariance to a *direct product* of groups $G_\Lambda \times S_C$, where the symmetry group of the lattice $G_\Lambda$ acts *across* unit cells and $S_C$, the symmetric group, acts *within* unit cells:

$$(g, h)\mathbf{h}_{(a,i)} = \mathbf{h}_{\left(\pi_g^S(a), \pi_h^U(i)\right)} \forall g \in G_\Lambda, h \in S_C. \tag{5}$$

In this way, differences in unit cell structures are dealt with by the symmetric group, where parameter-sharing can handle variable-sized inputs [72]. We still have to accommodate 14 different group actions $G_\Lambda$ corresponding to the different Bravais lattices. To avoid using a different model for each lattice, we propose two groups to deal with all the lattices. The first option is to consider the least symmetric Bravais lattice of primitive triclinic type and use its symmetry group $P\bar{1} = T_\Lambda \rtimes C_2$. This group is a subgroup of all the other lattice symmetry groups. The second option we consider is to simply use the symmetric group $S_\Lambda$ that is the symmetric group across unit cells, which is an overgroup of all the lattice symmetry groups. This leaves us with a hierarchy of groups, with $S_N$, the symmetric group over all atoms of a supercell, being the largest :

$$\underbrace{P\bar{1} \times S_C}_{P\bar{1}\text{-model}} \subseteq G_\Lambda \times S_C \subseteq \underbrace{S_\Lambda \times S_C}_{S_\Lambda\text{-model}} \subseteq S_N. \tag{6}$$

In our experiments, we use the two groups in this hierarchy for different levels of expressivity.

**Equivariant message passing** Having defined the group action, we can now build the Equivariant Crystal Network (ECN). We will seek to use the message passing framework, which has demonstrated good performance on molecular data, and generalize it to obtain equivariance to other groups than the symmetric group. The update equations for message passing framework are

$$\mathbf{m}_{ij} = \phi_e \left( \mathbf{h}_i^t, \mathbf{h}_j^t, \mathbf{e}_{ij} \right), \tag{7}$$

$$\mathbf{m}_i = \sum_{j \in N_i} \mathbf{m}_{ij},$$

$$\mathbf{h}_i^{t+1} = \phi_h \left( \mathbf{h}_i^t, \mathbf{m}_i^{t+1} \right).$$

The idea is to define parameter-sharing patterns for functions $\phi_e$ and $\phi_h$, such that there can be multiple versions while still retaining equivariance. Following [51], we first define the *parameter-sharing pattern* of the set of input nodes $\mathbb{N}$, with respect to group $G$ as the colored bipartite graph $\Omega \equiv (\mathbb{N}, \alpha, \beta)$, with the edge-color function $\alpha : \mathbb{N} \times \mathbb{N} \to \{1, \ldots, C_e\}$ and node-color function $\beta : \mathbb{N} \to \{1, \ldots, C_h\}$. We also consider the action of the group $G$ on edges $\Omega$ as $g \cdot (i, j) \doteq (\pi_g(i), \pi_g(j)) \, \forall g \in G$. We define the orbit $G \cdot (i, j)$ of edge $(i, j)$ as the set of edges in which it can be moved to by the group action : $G \cdot (i, j) \doteq \{g \cdot (i, j) \mid g \in G\}$. A similar definition applies to the orbit of a node, $G \cdot i \doteq \{\pi_g(i) \mid g \in G\}$.

We then make the following claim:

**Claim 6.1** *The layer defined by the $C_e$ functions $\phi_e^{\alpha(i,j)}$ and the $C_n$ functions $\phi_h^{\beta(i)}$*

$$\mathbf{m}_{ij} = \phi_e^{\alpha(i,j)} \left( \mathbf{h}_i^t, \mathbf{h}_j^t, \mathbf{e}_{ij} \right), \tag{8}$$

$$\mathbf{m}_i = \sum_{j \in N_i} \mathbf{m}_{ij}, \tag{9}$$

$$\mathbf{h}_i^{t+1} = \phi_h^{\beta(i)} \left( \mathbf{h}_i^t, \mathbf{m}_i^t \right), \tag{10}$$

*is G-equivariant if the parameter-sharing pattern $\Omega$ respects the equivariance condition:*

$$\alpha(i, j) = \alpha(k, l) \iff (k, l) \in G \cdot (i, j), \tag{11}$$

$$\beta(i) = \beta(j) \iff j \in G \cdot i. \tag{12}$$

The proof of this claim follows in Appendix A.2. In words, the group action on the graph creates node and edge orbits, and we use a different copy of $\phi_e$ and $\phi_h$ for each edge and node orbit, respectively. The computational process for producing the pattern is to find the orbit of G-action on the edges (nodes) [51], and the computational cost of this orbit-finding process grows linearly with the number of edges (nodes) [30].

This layer generalizes both MPNNs and equivariant multilayer perceptrons, such as CNNs. The MPNN is recovered with $G = S_n$ and a standard CNN with circular convolution with $G = T_\Lambda$, $\phi_e^{\alpha(i,j)} \left( \mathbf{h}_i^t, \mathbf{h}_j^t, \mathbf{e}_{ij} \right) = \mathbf{w}^{\alpha(i,j)} \cdot \mathbf{h}_i^t$ and $\phi_h^{\beta(i)} \left( \mathbf{h}_i^t, \mathbf{m}_i \right) = \text{ReLU} \left( \mathbf{m}_i^{t+1} + \mathbf{b}^{\beta(i)} \right)$

We now consider a product group $G \times H$, acting according to Eq. (5). From Claim 1 of [64], the equivariant linear map for this group is the Kronecker product of equivariant maps for individual groups; see also [45]. The reformulation for parameter-sharing patterns is the following. If parameters-sharing patterns $\Omega_1$ and $\Omega_2$ satisfy the equivariance condition for $G$ and $H$ respectively, then the parameter-sharing pattern $\Omega = (\mathbb{N} \times \mathbb{M}, \alpha, \beta)$ satisfies the equivariant condition if

$$\alpha : \mathbb{N} \times \mathbb{M} \times \mathbb{N} \times \mathbb{M} \to \{1, \ldots, C_{e,1}\} \times \{1, \ldots, C_{e,2}\},$$

$$\alpha(a, i, b, j) = (\alpha_1(a, b), \alpha_2(i, j)),$$

and

$$\beta : \mathbb{N} \times \mathbb{M} \to \{1, \ldots, C_{h,1}\} \times \{1, \ldots, C_{h,2}\}, \tag{13}$$

$$\beta(a, i) = (\beta_1(a), \beta_2(i)). \tag{14}$$

This simply means that a new color is defined in the product pattern for each possible combination of colors in the original patterns. Example 6.1 demonstrates this idea with a simple example.

We use MLPs to build functions $\phi_e^{\alpha(i,j)}$ and $\phi_h^{\beta(i)}$. The functions used in the experiments are detailed in Appendix A.4. In addition, we add a weighting factor to the edge aggregation 9, as this as been shown to be beneficial by [70, 53]:

$$\mathbf{m}_i = \sum_{j \in N_i} e_{ij} \mathbf{m}_{ij}, \quad \text{where} \quad e_{ij} = \phi_a (\mathbf{m}_{ij}), \tag{15}$$

and $\phi_a$ is simply a linear layer. This change does not affect the equivariance of the model.

---

**Example 6.1** In our running example, the parameter-sharing pattern for the message passing is produced by the Kronecker product of the pattern for the $P2$ group (see Appendix A.3 for a similar example with the $P6m$ group) and the pattern for the symmetric group $S_2$ as shown in the figure below (Top left). However, we only need to keep the colors for which there is a corresponding edge (Top right). Message passing is used on the resulting edge/node colored graph where similar colored nodes and edges use the same functions in message passing.

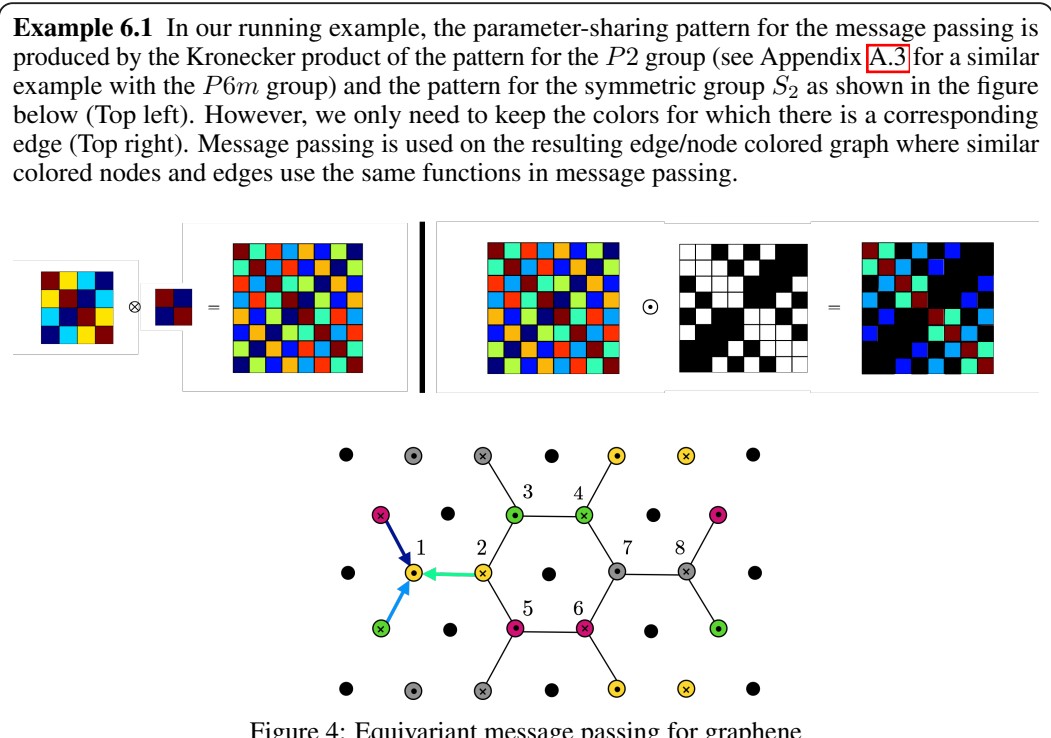

Figure 4: Equivariant message passing for graphene

---

## 7 Model implementation

**Network architecture** We aimed to keep the architecture of our ECN model as simple as possible. The model receives crystal structure graphs as input, with one-hot encoded feature vectors for each atom. We use 128-dimensional embeddings and keep the same dimension for hidden layers throughout the network. The ECN consists of 6 layers of the equivariant message passing operation 6.1. This is followed by a mean-pooling operation over node embeddings of each graph and a two-layer MLP that outputs the final prediction. Mean pooling is preferred to other options because we only predict *intensive* physical quantities. These properties are "per-atom" and do not depend on the choice of supercell. Therefore, selecting a pooling operation that respects this invariance makes sense. If we were to predict *extensive* quantities, sum pooling would be the preferred option.

**Input features** Many choices to encode atomic features are possible and have been suggested in the literature. We perform experiments over these variants. For each atom, we encode its type in a feature vector. We consider two strategies, using only information available from the periodic table. The first one is to use only the atomic number. The second one is to use the one-hot encoded group and period numbers for each atom. The second strategy could benefit from promoting better generalization since embeddings of atoms belonging to the same group or period will show a certain similarity. Around $24\%$ of the structures retained in the Materials Project dataset have been computed using the Hubbard-$U$ extension of DFT. Since this information can significantly influence the resulting properties [61] it is added to the atomic feature vector as a binary feature.

**Other details** We implemented our model using Pytorch [49]. We use a sparse implementation of the equivariant message-passing based on the Pytorch Scatter package [22]. We use the AdamW

optimizer [43] with weight decay regularization. The full hyperparameter setup is provided in Appendix A.7.

## 8 Experiments

**Materials Project**  We perform experiments using the Materials Project dataset [33] [1]. This standard dataset of materials informatics comprises more than 120K materials with a complete specification of their crystal structure and some important physical properties obtained with high-throughput DFT calculations.

The Materials Project dataset has not been initially built to serve as a machine learning benchmark; we perform some preprocessing to make it suitable. First, since multiple DFT calculations were sometimes performed on close initial configurations, some samples only show marginal structure differences and resulting properties. This is exemplified by the compound $Li_9Mn_2Co_5O_{16}$, which appears 322 times in the database. Such duplicates can result in training-test leakage. To prevent this, we consider two structures redundant if they have the same unit cell chemical formula and the same space group. Amongst a set of duplicate structures, the one with the lowest formation energy is chosen. Second, we filter out one-dimensional and two-dimensional materials from the dataset to only keep three-dimensional materials. Finally, we remove materials for which the unit cell contains more than 50 atoms. These are often associated with molecular and inorganic crystals with very different properties than the other materials. Training, validation, and test splits are 80%, 10%, and 10% of the dataset. We provide statistics on the processed dataset in Appendix A.6.

Following previous work, we predict a few relevant energetic properties: the formation energy $E$, the Fermi energy $E_F$, and the band gap $E_g$ for the subset of insulating materials. We also predict the binary insulator or conductor character material. Finally, we also predict the magnetic moment per atom $M$. This can be seen as a graph regression or classification task. For regression, training is performed using the mean-squared error (MSE) loss function, but we report the mean-absolute error (MAE). For classification, we use the cross-entropy loss function.

Table 1: Results on the Materials Project dataset.

| | Method | Property | | | | | |
|---|---|---|---|---|---|---|---|
| | | $E$(eV/atom) | $E_F$ (eV) | $M$ ($\mu_B$/atom) | $E_g$ (eV) | Metal precision | Nonmetal precision |
| Original | CGCNN | 0.039 | 0.363 | - | 0.388 | 80% | 95% |
| | MEGNET | 0.028 | - | - | 0.33 | 78.9% | 90.6% |
| | SCHNET | 0.041 | - | - | - | - | - |
| Ours | CGCNN | $0.048 \pm 0.0002$ | $0.307 \pm 0.001$ | $0.111 \pm 0.001$ | $0.399 \pm 0.006$ | $81.2\% \pm 3.0$ | $86.3\% \pm 3.0$ |
| | MEGNET | $0.056 \pm 0.0002$ | $0.365 \pm 0.007$ | $0.110 \pm 0.001$ | $0.434 \pm 0.006$ | $72.1\% \pm 3.0$ | $81.6\% \pm 4.0$ |
| | ECN-$P\bar{1}$ | $0.052 \pm 0.001$ | $0.303 \pm 0.004$ | $0.108 \pm 0.002$ | $0.44 \pm 0.02$ | $80\% \pm 4.0$ | $84\% \pm 4.0$ |
| | ECN-$S_\Lambda$ | $0.046 \pm 0.002$ | $0.281 \pm 0.007$ | $0.106 \pm 0.002$ | $0.390 \pm 0.02$ | $79.8\% \pm 2.0$ | $83.2\% \pm 1.0$ |

We compare the results obtained by our models to baselines [54, 70, 12]. Note that because these papers used different training and test splits and preprocessing schemes (even between themselves), our results cannot be directly compared. To alleviate that, we trained our own versions of two of the baselines using our splits and a similar training procedure. We obtain slightly better or comparable results to the baselines on all targets when evaluated on the same splits. The $S_\Lambda$ version offers better performance than the $P\bar{1}$ model overall, showing that it is more beneficial to lean on the side of having slightly more symmetry than necessary at the cost of some expressivity. We provide additional results for the model variants in Appendix A.8. The benefits of the increased expressivity on this task is in the not crucial, which we think can be explained in part by the relatively small size of the Materials Project dataset. In a larger data regime, we expect that the benefit of increased expressivity will outweight the cost in generalization capability.

**Perov-5**  Finally, we perform experiments using the Perov-5 dataset [9] as provided by [69]. In this dataset, all the materials share the same Perovskite crystal structure. The task considered is the regression of the heat of formation computed through DFT. Results are shown on Table 2.The improvement on this dataset is significantly more important for the proposed model compared to the

---

[1]We use version 2021.05 of the dataset

baselines than on the Materials Project dataset. We hypothesize that the fact that all the structures are shared in this dataset allows the model to specialize more efficiently leading to better generalization.

## Conclusion

We have shown how to leverage crystal symmetry to build more expressive and physically motivated neural networks for materials data. This allows us to obtain a close equivalent of group equivariant convolution on this data structure. These models show excellent accuracy in supervised property prediction, which supports the idea that symmetry is a useful inductive bias. Such models could be used for other tasks on materials such as dynamics prediction, if the dynamics approximately preserves the crystal structure. We also think that these models have significant potential on more abstract condensed matter systems such as spin models and free-fermion

| Method | Property Heat All |
|---|---|
| CGCNN | $0.047 \pm 0.000$ |
| MEGNET | $0.059 \pm 0.006$ |
| ECN-$S_\Lambda$ | $\mathbf{0.038 \pm 0.004}$ |

Table 2: Perov-5 results

models. We have also defined equivariant message passing, a generalization of the MPNN framework that can potentially be used on any data structures for which a group can capture the symmetry in sparse interactions between the basic elements. One limitation of this approach is that it is not clear how to handle structures with different groups without using a larger group like $S_\Lambda$. A potential solution is drawing inspiration from the Natural Graph Neural Networks framework introduced in [27]. Another area of future improvement is on the computational efficiency of the equivariant message passing, which does not benefit from optimized algorithms available for convolutions.

## Acknowledgments and Disclosure of Funding

We thank Mehran Shakerinava, Christopher Morris, Joey Bose, Simon Verret and the anonymous reviewers for their valuable comments. This project is in part supported by the CIFAR AI chairs program and NSERC Discovery. S.-O. K.'s research is also supported by IVADO and the DeepMind Scholarship. Computational resources were provided by Mila and Compute Canada.

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
