# A Appendix

Table 3: Bravais lattices in 2 and 3 dimensions.

| BRAVAIS TYPE | SYMMETRY GROUP | ABSTRACT POINT GROUP |
|---|---|---|
| 2 dimensions | | |
| Oblique | $P2$ | $C_2$ |
| Rectangular | $Pmm$ | $D_2$ |
| Centered rectangular | $Cmm$ | $D_2$ |
| Square | $P4m$ | $D_4$ |
| Hexagonal | $P6m$ | $D_6$ |
| 3 dimensions | | |
| Primitive triclinic | $P1$ | $C_2$ |
| Primitive monoclinic | $P2/m$ | $C_{2h}$ |
| Base-centered monoclinic | $C2/m$ | $C_{2h}$ |
| Primitive orthorhombic | $Pmmm$ | $D_{2h}$ |
| Base-centered orthorhombic | $Cmmm$ | $D_{2h}$ |
| Body-centered orthorhombic | $Immm$ | $D_{2h}$ |
| Face-centered orthorhombic | $Fmmm$ | $D_{2h}$ |
| Primitive tetragonal | $P4/mmm$ | $D_{4h}$ |
| Body-centered tetragonal | $C4/mmm$ | $D_{4h}$ |
| Rhombohedral | $R\bar{3}m$ | $D_{3d}$ |
| Hexagonal | $P6/mmm$ | $D_{6h}$ |
| Primitive cubic | $Pm\bar{3}m$ | $O_h$ |
| Body-centered cubic | $Im\bar{3}m$ | $O_h$ |
| Face-centered cubic | $Fm\bar{3}m$ | $O_h$ |

## A.1 Graph-building strategies

The graphs were built using the `IsayevNN` class from the `pymatgen` [48] package. It implements the commonly used Voronoi tessalation to define neighbors. Two atoms are considered bonded if they share a face in the Voronoi tessalation of the supercell and their distance is less than the sum of the atomic Cordero radii (a measure of the atomic radius) plus a cutoff $\Delta = 0.5$Å. This value of the cutoff was increase compared to [32] to reduce the number of disconnected graphs.

We provide statistics for the graphs obtained by the method described in Section 5. A hard cutoff on atomic distances of 6Å is also imposed on atomic distances.

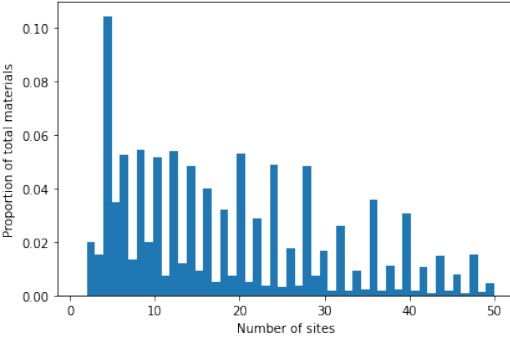

Figure 5: Histogram of the number of primitive cell sites per material in the processed Materials Project dataset.

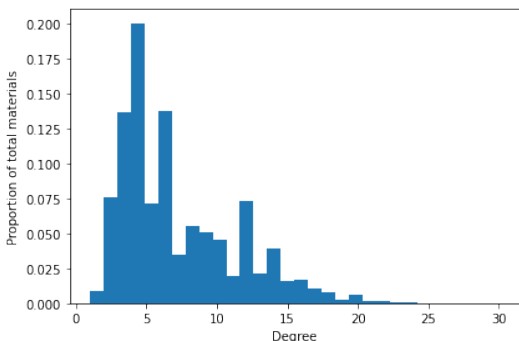

Figure 6: Degree distribution for all graphs.

## A.2 Proof of Claim 6.1

We prove that the function defined by

$$\mathbf{m}_{ij} = \phi_e^{\alpha(i,j)} \left( \mathbf{h}_i^t, \mathbf{h}_j^t, \mathbf{e}_{ij} \right), \tag{16}$$

$$\mathbf{m}_i = \sum_{j \in N_i} \mathbf{m}_{ij}, \tag{17}$$

$$\mathbf{h}_i^{t+1} = \phi_h^{\beta(i)} \left( \mathbf{h}_i^t, \mathbf{m}_i^t \right), \tag{18}$$

with

$$\alpha(i,j) = \alpha(k,l) \iff (k,l) \in G \cdot (i,j), \tag{19}$$

$$\beta(i) = \beta(j) \iff j \in G \cdot i, \tag{20}$$

is $G$-equivariant by proving the equivariance of every step and using the fact that function composition preserves equivariance.

First, we need to show equivariance of the message function on Equation (16)

$$\mathbf{m}_{\pi_g(i),\pi_g(j)} = \phi_e^{\alpha(i,j)} \left( \mathbf{h}_{\pi_g(i)}^t, \mathbf{h}_{\pi_g(j)}^t, \mathbf{e}_{\pi_g(i),\pi_g(i)} \right) \forall g \in G. \tag{21}$$

Using Equation (19), we have

$$\phi_e^{\alpha(i,j)} \left( \mathbf{h}_{\pi_g(i)}^t, \mathbf{h}_{\pi_g(j)}^t, \mathbf{e}_{\pi_g(i),\pi_g(i)} \right) = \phi_e^{\alpha(\pi_g(i),\pi_g(j))} \left( \mathbf{h}_{\pi_g(i)}^t, \mathbf{h}_{\pi_g(j)}^t, \mathbf{e}_{\pi_g(i),\pi_g(i)} \right) \forall g \in G. \tag{22}$$

The right-hand side is equal to $\mathbf{m}_{\pi_g(i),\pi_g(j)}$ by definition.

The message aggregation step at Equation (17) is permutation equivariant

$$\mathbf{m}_{\pi_g(i)} = \sum_{\pi_g(j) \in N_{\pi_g(i)}} \mathbf{m}_{\pi_g(i),\pi_g(j)}. \tag{23}$$

Finally, we need to show that the node function on Equation (18) is also equivariant

$$\mathbf{h}_{\pi_g(i)}^{t+1} = \phi_h^{\beta(i)} \left( \mathbf{h}_{\pi_g(i)}^t, \mathbf{m}_{\pi_g(i)}^t \right) \forall g \in G. \tag{24}$$

Using Equation (20), we find

$$\phi_h^{\beta(i)} \left( \mathbf{h}_{\pi_g(i)}^t, \mathbf{m}_{\pi_g(i)}^t \right) = \phi_h^{\beta(\pi_g(i))} \left( \mathbf{h}_{\pi_g(i)}^t, \mathbf{m}_{\pi_g(i)}^t \right) \forall g \in G. \tag{25}$$

The right-hand side is equal to $\mathbf{h}_{\pi_g(i)}^{t+1}$ by definition.

### A.3 Parameter sharing patterns

The parameter sharing patterns are computed using a group-theoretical orbit finding algorithm that has linear complexity in the number of size of the generating sets of a group and in the number of edges in the coloured bipartite graphs [29, 56].

We show the parameter sharing patterns for the different Bravais lattice groups (Figure 7 and Figure 8). For the 2-dimensional groups, we use a $2 \times 2$ supercell and for the 3-dimensional groups a $2 \times 2 \times 2$ supercell. Note that the numbering of the unit cells within the supercell is chosen by convention and can vary for different lattices.

Notice that the parameter-sharing patterns for different groups can be the same. This is because, for different groups, the group action can induce the same orbits on the bipartite graph. In particular, for some groups (Figure 7c), the pattern collapses to that of the symmetric group. This can be undesirable since it reduces expressivity. This can be alleviated by using larger supercells or eliminated by considering the group acting on itself instead of on supercells, as done in [13].

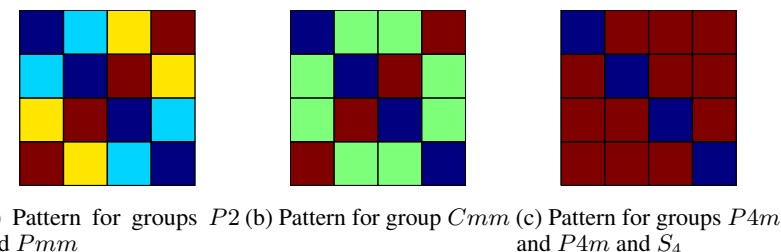

(a) Pattern for groups $P2$ and $Pmm$ (b) Pattern for group $Cmm$ (c) Pattern for groups $P4m$ and $P4m$ and $S_4$

Figure 7: Parameter sharing patterns for 2-dimensional Bravais lattices groups

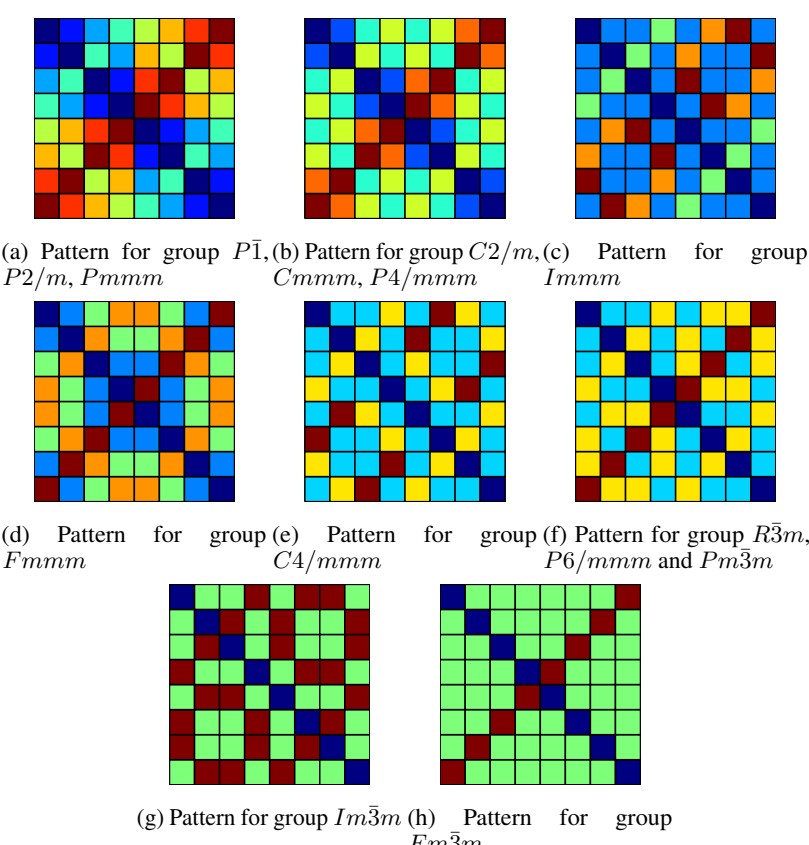

(a) Pattern for group $P\bar{1}$, $P2/m$, $Pmmm$ (b) Pattern for group $C2/m$, $Cmmm$, $P4/mmm$ (c) Pattern for group $Immm$

(d) Pattern for group $Fmmm$ (e) Pattern for group $C4/mmm$ (f) Pattern for group $R\bar{3}m$, $P6/mmm$ and $Pm\bar{3}m$

(g) Pattern for group $Im\bar{3}m$ (h) Pattern for group $Fm\bar{3}m$

Figure 8: Parameter sharing patterns for 3-dimensional Bravais lattices groups

## A.4 Node and edge functions details

For the node functions $\phi_h^{\beta(i)}$, implementation is facilitated by the fact that for all the group we consider, the group action is transitive. Therefore, for all $i, j \in \mathbb{N}$, $j \in G \cdot i$. This implies that for all $i, j \in \mathbb{N}$, $\beta(i) = \beta(j)$. There is thus only one node function, which we implement with two-layer MLP with a residual connection.

For the edge functions $\phi_h^{\alpha(i,j)}$, we do not explicitly build the parameter-sharing patterns. This would be computationally expensive in a dataset in which sample have different unit cell sizes like Materials Project because it would require to build patterns for each unit cell size. Instead, we use an approach inspired by [45]. Let the group be $G_\Lambda \times S_C$ and each node be represented by the embedding $\mathbf{h}_{(a,i)}$ where the first index encodes the unit cell within the supercell and the second index encodes the identity of the atom within the unit. Using Theorem 1 of [45], we can define the edge function as

$$\phi_h^{\alpha((a,i),(b,j))}\left(\mathbf{h}_{(b,j)}^t, \mathbf{h}_{(b,j)}^t, \mathbf{e}_{(a,i),(b,j)}\right) \tag{26}$$
$$= \delta_{a,b}\phi_1^{\alpha_{G_\Lambda}(i,j)}\left(\mathbf{h}_{(b,j)}^t, \mathbf{h}_{(b,j)}^t, \mathbf{e}_{(a,i),(b,j)}\right) + \phi_2^{\alpha_{G_\Lambda}(i,j)}\left(\mathbf{h}_{(b,j)}^t, \mathbf{h}_{(b,j)}^t, \mathbf{e}_{(a,i),(b,j)}\right),$$

where $\alpha_{G_\Lambda}$ is obtained from the parameter-sharing pattern of the group $\alpha_{G_\Lambda}$, which is shared for all the dataset.

In our implementation, $\phi_1^{\alpha_{G_\Lambda}(i,j)}$ and $\phi_2^{\alpha_{G_\Lambda}(i,j)}$ are built explicitly for as two-layer MLPs for all possible values of $\alpha_{G_\Lambda}(i,j)$.

## A.5 $E(n)$-equivariant version of ECN

Our architecture can be made $E(n)$-equivariant instead of $E(n)$-invariant. The basic idea is that multiple edge functions can always be introduced using the parameter sharing pattern. Using a simple generalization of the EGNN model [53], we can define the following layer

$$\mathbf{m}_{ij} = \phi_e^{\alpha(i,j)}\left(\mathbf{h}_i^t, \mathbf{h}_j^t, \left\|\mathbf{x}_i^l - \mathbf{x}_j^l\right\|, \mathbf{e}_{ij}\right), \tag{27}$$
$$\mathbf{x}_i^{l+1} = \mathbf{x}_i^l + C\sum_{j \neq i}\left(\mathbf{x}_i^l - \mathbf{x}_j^l\right)\phi_x^{\alpha(i,j)}\left(\mathbf{m}_{ij}\right)$$
$$\mathbf{m}_i = \sum_{j \in N_i}\mathbf{m}_{ij},$$
$$\mathbf{h}_i^{t+1} = \phi_h^{\beta(i)}\left(\mathbf{h}_i^t, \mathbf{m}_i^{t+1}\right).$$

## A.6 Materials Project dataset

We hereafter report the number of samples in the Materials Project dataset at different levels of the preprocessing scheme :

**(a)** Full dataset

**(b)** No duplicates

**(c)** No duplicates and unit cell size constraint

**(d)** No duplicates, unit cell size constraint and 3D materials

**(e)** No duplicates, unit cell size constraint, 3D materials and valid graphs

**(f)** Insulators, no duplicates, unit cell size constraint, 3D materials and valid graphs

Table 4: Number of entries in the Materials Project dataset with processing

| (a) | (b) | (c) | (d) | (e) | (f) |
|---|---|---|---|---|---|
| 126126 | 114605 | 96315 | 82229 | 78649 | 33971 |

We also report the mean and standard deviation of each target for the processed dataset

Table 5: Targets

| Property | $E$(eV/atom) | $E_F$ (eV) | $M$ ($\mu_B$/atom) | $E_g$ (eV) |
|---|---|---|---|---|
| **Mean** | -1.42 | 3.77 | 0.16 | 2.02 |
| **Std. dev.** | 1.13 | 2.63 | 0.44 | 1.56 |

## A.7 Hyperparameters

We use the same training setup for all the models. The learning rate is initialized at $1 \times 10^{-3}$ with a scheduler halving it after each 25 epoch plateau on the validation loss. Training is perfomed for 1000 epochs or until the learning rate reaches $1 \times 10^{-6}$. We performed sweeps over learning rates for all the models to verify that this is indeed a setup in which all models train well.

For the ECN models, we optimized the number of layers, the embedding dimension, the weight decay parameter and dropout (without noticing significant improvement). We use 6 layers of message passing operations. The $\phi_e^{\alpha(i,j)}$ and $\phi_h^{\beta(i)}$ functions are 1-hidden layer MLPs. For the $\phi_e^{\alpha(i,j)}$ functions, the weights for the hidden layer and output layers are shared across all $\alpha(i,j)$. This was found to perform slightly better. The SWISH activation function is used. The feature dimension of node embeddings is 100 across all the network. For edge embeddings, the dimension is set at 20.

For the CGCNN model, we used the same embedding sizes (they were not specified in the original paper) and 2 convolution layers. For MEGNet, we used the same architecture setup as in the original paper.

## A.8 Supplementary results

| Method | Property | | | | | |
|---|---|---|---|---|---|---|
| | $E$(eV/atom) | $E_F$ (eV) | $M$ ($\mu_B$/atom) | $E_g$ (eV) | Metal precision | Nonmetal precision |
| ECN-$S_\Lambda$ Hubbard | 0.052 | 0.290 | 0.109 | 0.368 | 81.2% | 82.9% |
| ECN-$S_\Lambda$ Group+Period | 0.051 | 0.295 | 0.109 | 0.386 | 77.6% | 85.6% |

Table 6: Results for model variants on Materials Project