# OpenReview forum: "Equivariant Networks for Crystal Structures"
_NeurIPS.cc/2022/Conference — NeurIPS 2022 Accept_

### Official Review · Reviewer_os5D · 2022-07-09

**Rating:** 7
**Confidence:** 3
**Soundness:** 3 good
**Presentation:** 4 excellent
**Contribution:** 3 good

**Summary:**

This paper studies bespoke equivariant neural architectures for processing crystal structures. Firstly, it is observed that crystals are discrete structures that exhibit substantially richer symmetries than the permutation group (which would normally used for abstract graph representations), which might allow for relaxed equivariance conditions, and hence, stronger architectures. Additionally, the authors show how to combine symmetry groups coming from various crystal structures within the same dataset, through the use of carefully constructed products of groups. The resulting equivariant architectures show promising results on a standard material science benchmark.

**Questions:**

Q1. Could you comment on whether there exist any other suitable baselines and/or benchmark datasets you could have used, provided you had more computational power at your disposal?

Q2. Would you give any "items of future work" for interested authors who do have access to more computation?

Q3. Could you provide at least 1-2 additional seeds for some of the models you evaluated, to get a feel for the variance?

Q4. Does this paper, to the best of your knowledge, present the first rigorous exploration of geometric DL over crystal inputs, or are there any pieces of related work you would highlight? Some related works are already listed in the paper, but it is unclear if any of them are mindful of the symmetry structures you point out.

**Limitations:**

No concerns.

**Strengths And Weaknesses:**

**Originality**:
From the methodological point of view, the proposed architectures and theoretical treatment appear like a direct application of the principles of geometric deep learning to the specific case of crystal symmetries. The idea of using the direct product of symmetry group is also in line with the GDL principles; see, for example, "An equivariant Bayesian convolutional network predicts recombination hotspots and accurately resolves binding motifs" from Brown & Lunter, which uses similar ideas to construct reverse-complement-symmetric architectures (for the processing of DNA).

I do not have significant background in material science applications, therefore I cannot say for sure to what extent such ideas have already been explored in the context of crystals. However, the authors' aim to bring such symmetries into the spotlight, especially in light of naïve applications of graph neural networks to such problems, are certainly remarkable and deserve credit in either case.

**Quality**:
I find the proposed model to be solidly theoretically grounded, and the results to be reasonable and showing benefits against relevant GNN baselines. Of course, as before, I may not have sufficient context on what are the most meaningful architectures in this domain to use.

**Clarity**:
This is one of the paper's absolute strongpoints. The use of guided examples (particularly in the case of graphene) are remarkably well-made and illustrate the points the paper makes perfectly. The paper was a delight to read as a result.

**Significance**:
The main concern for me in the authors' evaluation is the lack of error bars, which the authors stated they are unable to provide due to computational resources. Would it be possible at least to get 1-2 additional seeds for the authors' proposal, in order to get some sense of the variance of the results? Otherwise, the results appear significant, but only one benchmark dataset is used (albeit, one that seems to be a standard choice for materials science).

**Overall**:
When weighing in on the different aspects of this paper's presentation, I tend to be in favour of accepting it. It presents a great application area for geometric deep learning, a principled investigation of how GDL can be applied in this space, and all of this is presented in a way that is very likely to reduce the barrier of entry into the field. I look forward to the release of the dataset and other artefacts.

---

> ### Author Response · Authors · 2022-08-02
> **Response**
>
> We thank the reviewer for this thoughtful review and we are glad to see their positive assessment. We have updated the manuscript to accommodate their feedback; in particular we have added the error bars that were previously omitted due to computational constraints.
>
> Q : Could you comment on whether there exist any other suitable baselines and/or benchmark datasets you could have used, provided you had more computational power at your disposal?
>
> A : We think that the open catalyst dataset [10] would be an excellent candidate to test this model. First, the materials in this dataset are naturally represented by more than only one unit cell, eliminating the need to create 2x2x2 supercells. Second, it is much larger than Materials Project. We can expect that in this regime, the increased expressivity would prove more beneficial. Experimenting with this dataset was outside the scope of this work as it comes with additional challenges in terms of change of prediction task/architecture, preprocessing, and the need for significant computational resources due to its size.
>
> Q : Would you give any "items of future work" for interested authors who do have access to more computation?
> A : This is a great question that we have further discussed in the revised manuscript. First, we think that this approach to build models could be used with bigger datasets (like OC20 mentioned above as a real test). Second, these models would be very suitable for studies of single materials, as opposed to datasets of very diverse materials. For practical applications, researchers are often interested in fitting the potential energy of a single material using different configurations of the atomic positions [80, 81, 82]. We think that our model could perform well in this setting because the diversity of structures would not be an obstacle to generalization. Third, we think that this approach could be applied to condensed matter physics on spin lattices [74, 75, 76] and lattice fermion models [77], for which coordinates and the euclidean group are not relevant. Finally, note that the equivariant message passing approach can be applied to other types of structured graphs (for example trees) as long as a structure preserving group can be defined on the graph.
>
> Q : Could you provide at least 1-2 additional seeds for some of the models you evaluated, to get a feel for the variance?
> A : Yes, we agree and have performed additional runs to address this. We report an estimate of the standard deviation on the MAE.
>
> Q : Does this paper, to the best of your knowledge, present the first rigorous exploration of geometric DL over crystal inputs, or are there any pieces of related work you would highlight? Some related works are already listed in the paper, but it is unclear if any of them are mindful of the symmetry structures you point out.
> A : We are unaware of other work that applies these ideas to crystal structures rigorously. In particular, using crystal symmetries in the corresponding deep model is a natural choice that we have not seen in prior work. We also believe our approach in bridging graph neural networks and equivariant architectures can be of interest beyond application to crystals.
>
> [80] S. A. Ghasemi, A. Hofstetter, S. Saha, and S. Goedecker. Interatomic potentials for ionic systems with density functional accuracy based on charge densities obtained by a neural network. Phys. Rev. B, 92:045131, Jul 2015.
> ​​[81] R. Kobayashi, D. Giofr ́e, T. Junge, M. Ceriotti, and W. A. Curtin. Neural network potential for al-mg-si alloys. Phys. Rev. Materials, 1:053604, Oct 2017.
> [82] J. Behler. Four generations of high-dimensional neural network potentials. Chemical Reviews, 121(16):10037–10072, 2021.

---

> > ### Comment · Reviewer_os5D · 2022-08-03
> > **Rebuttal acknowledged**
> >
> > Thank you for your comments, and your efforts to offer variance estimates for your models as well as additional datasets. This is greatly appreciated.
> >
> > I will retain my recommendation, and will support this paper for acceptance. Good luck!

---

### Official Review · Reviewer_PzGa · 2022-07-13

**Rating:** 4
**Confidence:** 2
**Soundness:** 3 good
**Presentation:** 2 fair
**Contribution:** 2 fair

**Summary:**

This paper generalizes the message passing operation which preserve $S_n$ equivariant to more general permutation groups, specifically crystalline symmetry groups, by parameter sharing pattern respecting equivariance condition. Models with the proposed message passing operation achieve comparable results with SOTA on the Materials Project dataset.

The idea is to improve the expressivity of GNNs by building an 8-time larger supercell and imposing parameter sharing using space group symmetry operations.

It takes me quite long time to review this paper, partly because I have less background of Crystal. I think this manuscript would be suitable for publication on journals like JCTC, rather than a machine learning conference.

**Questions:**

+ I am still a little confused what the authors mean by P1 and Sp symmetry group exactly. This is the symmetry of the P1 group: http://img.chem.ucl.ac.uk/sgp/large/002az1.htm. I don’t know what the authors mean by Sp group, which is not in the space group table. Example 6.1 is for a P2 group, which is different from P1 and Sp group.
- For the experiments, the authors created a new dataset with 33971 crystals. They assume two structures are redundant if they have the same space group and chemical composition. This is problematic because different crystal structures can have both the same group and chemical composition.
- For data preprocessing in Appendix A4, motivation of the last step is not clear. Why do they only include insulators?
- The improvements with respect to SchNet and CGCNN are relatively small. The author didn’t consider latest methods like ALIGNN. Matbench provides a comparison between different models which could be used as a benchmark dataset (https://matbench.materialsproject.org/)

**Limitations:**

See above.

**Strengths And Weaknesses:**

Strength:
+ The motivation that full permutation equivariance might be overly restrictive is insightful and interesting.
+ The empirical result is interesting.

Weakness:
+ The presentation is unclear. The paper could be more self-contained if “Voronoi face” and “Cordero radii” are explained in the main text or appendix. In line 199-209, there should be definition of $S_N$, $S_P$, $S_N$ and $S_{P×N}$. It seems that $S_N$ means permutations within unit cell, $S_P$ means permutations across unit cells (in supercell) derived from point group $P$, and $S_{P×N}$ means all permutations?
+ Experiments are only conducted on one dataset, using different training and test splits and preprocessing scheme from previous works. The improvement is marginal according to line 301-302: “The $S_P$ version offers **slightly** better performance than the $P\bar{1}$ model overall” – the differences comparing to previous work are similar to the different between two proposed variants in Table 1.
+ As line 302-303 suggest, having more symmetry than necessary might be beneficial. The absence of ECN-“full permutation equivariance” in experiment drastically weaken the empirical result, i.e. the result cannot support the core proposition of this paper, that generalized permutation equivariance is better than full permutation equivariance (both input supercell) for crystal data.
+ Novelty of the proposed method is limited. The proposed method could be viewed as an application of [49] Equivariance Through Parameter-Sharing for crystal symmetry group.

---

> ### Author Response · Authors · 2022-08-02
> **Response**
>
> We thank the reviewer for their feedback and hope to address all of their questions and concerns below. We have also revised the manuscript to address these issues.
>
> Q: The paper could be more self-contained if “Voronoi face” and “Cordero radii” are explained in the main text or appendix.
> A:  We have added more details on this in the appendix. The reason for the initial omission was that this terminology is common in the literature of applications of machine learning to materials [4, 30, 68 74].
>
> Q : There should be explicit definitions of the different groups
> A : We have included detailed definitions in the updated manuscript.
>
> Q : The improvements with respect to SchNet and CGCNN are relatively small. The author didn’t consider latest methods like ALIGNN.
> A : We agree that the improvement with respect to baselines is marginal. We have performed an experiment on a new dataset for which improvement is more significant. In addition, we have material in the revised manuscript to nuance the results.
>
> Q : For data preprocessing in Appendix A4, motivation of the last step is not clear. Why do they only include insulators?
> A : This subset of the dataset is only used for band gap regression. The reason is that for conductors, the band gap is zero. This is the common approach, also used in the baselines. For all the other tasks, the dataset with 78649 samples is used.
>
> Q : For the experiments, the authors created a new dataset with 33971 crystals. [Assuming that structures with the same chemical composition and space group are redundant] is problematic because different crystal structures can have both the same group and chemical composition.
> A : The dataset actually includes 78649 samples, as mentioned above. We think it is much better to remove duplicate structures, even if accidentally removing a few valid structures, than not doing so. Using a more stringent approach (directly comparing atomic positions) would be prohibitively computationally expensive. We have examined the removed structures to validate this approach.
>
> [79] S. P. Ong, W. D. Richards, A. Jain, G. Hautier, M. Kocher, S. Cholia, D. Gunter, V. L. Chevrier, K. A. Persson, and G. Ceder. Python materials genomics (pymatgen): A robust, open-source python library for materials analysis. Computational Materials Science, 68:314 – 319, 2013.

---

### Official Review · Reviewer_MuHL · 2022-07-22

**Rating:** 6
**Confidence:** 4
**Soundness:** 3 good
**Presentation:** 3 good
**Contribution:** 3 good

**Summary:**

The work introduces a method to design neural networks that are equivariant w.r.t. crystal symmetries, which arise in the description of materials structures. Given the success of symmetry as an inductive bias in molecular and materials science, based so far mostly on the groups SO(3), E(3), and the permutation group, the authors hypothesise that explicitly "baking in" equivariance w.r.t. crystal symmetries may provide a useful inductive bias for machine learning on crystals. First an introduction is given to a few basic concepts of solid-state physics such as lattices, point groups, and space groups based on the graphene example. Then the equivariant network architecture is described as a form of equivariant message passing. Equivariance is achieved by defining a weight-tying pattern that respects the symmetry of the crystal structure. Finally, the method is demonstrated on a series of targets available from Materials Project data. The method performs favorably.

**Questions:**

1. "In this paper, we address the problem of designing equivariant layers for supervised learning on materials." - why is this a problem? It is not clear a prior why equivariance (as opposed to invariance) is a desirable property as most target properties are invariant under the prescribed symmetries.

2. The introduction could strongly benefit from some description of when crystal symmetries are important. E.g. there are many applications in computational materials science where crystal symmetry is not helpful (e.g. molecular dynamics, relaxations, ... ), but only the symmetries of E(3) as well as the symmetric group are still valid. On the other hand, current methods can *only* incorporate these and would not properly exploit additional symmetry information as proposed here. This should be laid out more clearly.

3. Typo: "tilling the space" should likely be "filling the space"?

4. I wonder if it might be beneficial to use a Bessel embedding instead of a Gaussian one? A number of works in the ML potential field have adopted this and it's been found to give improved performance, see Klicpera, J., Groß, J., & Günnemann, S. (2020). Directional message passing for molecular graphs. arXiv preprint arXiv:2003.03123. for example.

5. "This has the benefit of simplicity while still allowing a complete description of the input structure" --> it's not clear why a Gaussian encoding vs just the interatomic distance is simpler, this sentence should be clarified. Distance is equally complete + simple, if those were the only 2 criterions (I realize it performs better in practice, but these are not the reasons why).

6. Section 6.2 would benefit from some more detail. A lot of time is (rightfully) spent on group theory of crystals but then the actual core idea is introduced only briefly. This could be explained in more detail and ideally also giving some theoretical background. In particular, it is not clear how exactly the parameter-sharing is implemented: there is this sentence "For our model, we choose functions \phi_e^{\alpha(i, j)} and \phi_h^{\beta(i, j)} to be MLPs with one hidden layer", but it is not clear whether there is one separate MLP for each possible pair (i, j)? The appendix further confuses this by stating that "the weights for the hidden layer and output layers are shared across all \alpha(i, j)" --> this presentation would *strongly* benefit from more detail, in particular how the parameter-sharing is implemented.

7. In the comparison to other methods on MP: it is great that the authors do a proper filtering of the MP structures, this is solid data cleaning. However, when re-training another group's model there is often a chance that one underreports the performance of that model b/c the authors will have a better understanding of their own model's hyperparameters and a stronger incentive to optimize it. While this is in some sense difficult to avoid it would be great if the authors could at least clarify how the hyperparameters for the competing models were defined and if any hyperparameter optimization was performed and if yes, how that compared to the hyperparameter optimization done for ECN.

8. The empirical results show only a small improvement. Given that no estimate of the statistical variance of the MAEs is given, it's difficult to say if these improvements are even statistically significant. If computationally feasible, this would *strongly* benefit from that. In addition, it would be great if the authors could add a (albeit speculative) sentence on why a model that explicitly accounts for crystal symmetry only marginally outperforms models that don't.

9. In the conclusion the authors say "Such models could be used for other tasks on materials such as dynamics prediction [...]" --> this is difficult to imagine since any crystal symmetry would immediately be gone under molecular dynamics? Or was something different meant here? If yes, please specify more clearly.

10. The authors at some point outline 2 strategies for encoding atoms, one based simply on a one-hot of the atomic number, one based on row+column of the PT and on a binary +U feature -- I could not find these experiments anywhere? Did I miss them?


**Strengths And Weaknesses:**

Strengths: The paper provides an elegant solution to a non-trivial problem with potential applications in materials science. The solution to this problem is non-trivial and clean. The presentation of the mathematics as well as the required background in solid-state physics / crystal symmetry is well-developed using the classic graphene text book example and understandable even for an audience without a physics/materials science background. The experiment is being done with great care w.r.t. data set selection.

Weaknesses: The paper only shows a single, simple experiment reporting average metrics on a common benchmark data set. While it shows marginal improvements over existing methods, the method provides no additional insights into specific materials or an attempt to discover new ones. It is not tested *in the wild* on a real materials problem. There is also no effect undertaken into understanding *why* the method works. Design choices are not tested or compared to others. The work would also strongly benefit from analyzing the inference time required on these networks. In particular the choice to construct a 2x2x2 supercell seems like it could rapidly lead to high inference cost. This leaves open many questions about whether such a system has potential for real-world impact. Finally, the presentation of the actual model leaves open many question. It is not immediately apparent *how* exactly the parameter-sharing MLPs are implemented.

---

> ### Author Response · Authors · 2022-08-02
> **Response**
>
> We wish to thank the reviewer for this helpful and comprehensive referee report, and we are glad to see their positive assessment of our methodology and presentation. Below we answer the questions and explain how the revised manuscript accommodates this detailed and useful feedback.
>
> Q : “While it shows marginal improvements over existing methods, the method provides no additional insights into specific materials or an attempt to discover new ones.”
> A : We agree that the improvement with respect to baselines is marginal. We have performed an experiment on a new dataset for which improvement is more significant. In addition, we have material in the revised manuscript to nuance the results.
>
> Q : The work would also strongly benefit from analyzing the inference time required on these networks.
> A : We agree with your suggestion. This will be added in the final version of the paper as due to time constraint this could not be done for the revision.
>
> Q : Finally, the presentation of the actual model leaves open many questions. It is not immediately apparent how exactly the parameter-sharing MLPs are implemented.
> A : Thank you for this comment, we have modified the manuscript to include more details on the implementation of the parameter-sharing patterns.
>
> Q : It is not clear a prior why equivariance (as opposed to invariance) is a desirable property as most target properties are invariant under the prescribed symmetries.
> A : We have elaborated on this statement in the revised paper. We see mainly two reasons why it is important to consider equiariance and not only invariance. First, even for an invariant prediction task, the model will be much more expressive if the intermediate layers are equivariant instead of invariant. This is analogous to CNNs where we use intermediate equivariant layers and only the ouput layer is invariant. The second reason is that equivariant models can be used to predict local quantities (for example, local magnetization [29], charge distribution [73], forces per atom).
>
> Q : The introduction could strongly benefit from some description of when crystal symmetries are important.
> A : We agree and we will include a brief discussion on this.
>
> Q : On the question of raw distance encodings vs Gaussian encodings vs Bessel encodings
> A : We have clarified this issue. Indeed, raw distance encodings are also complete, we did not mean to say that Gaussian embeddings are better on that side. But we decided to use Gaussian embeddings to be consistent with what the other baselines use and avoid introducing another source of variation.
>
> Q : Hyperparameter optimization for baseline and proposed model
> A : We have added more detail in the appendix with regard to the hyperparameter optimization on the baseline models; it was done for the learning rate only. Note that the only difference between our experiment and the baselines experiments is that they have cleaned the data and used different splits. The baselines are also not consistent between themselves with regard to that. We expect that such changes in data cleaning and splitting should not significantly affect hyperparameters.
>
> Q : It would be great if the authors could add a (albeit speculative) sentence on why a model that explicitly accounts for crystal symmetry only marginally outperforms models that don't.
> A : Thank you for this suggestion. Based on the evidence from our experiments on perov-, where we have a dataset of crystals that share the same structure, we have a reasonable explanation: in the materials project dataset, the same model is applied to various crystal structures, which prevents its specialization. Here, we only see a marginal improvement over the baselines. However, in the perov-2 dataset, we see a more significant improvment, as expected.
> We have added this analysis to the revised manuscript.
>
> Q : [...] no estimate of the statistical variance of the MAEs is given
> A : We have performed additional runs to address this. We report an estimate of the standard deviation on the MAE (lack of error bars at the time of submission was due to computational constraints and it is addressed now.)
>
> Q : [Dynamics prediction] is difficult to imagine since any crystal symmetry would immediately be gone under molecular dynamics?
> A : We are thinking about cases where the crystal structure is approximately preserved under dynamics (for example in catalyst-absorbate reactions [10]). In this case, the model will correctly approximate a symmetric function even if the input is asymmetric. This is analogous to CNNs in computer vision: images, as projections of 3D world, are not exactly translationally symmetric, yet using this inductive bias proves useful despite its inaccuracy. We have clarified this in the manuscript.
>
> Q : The authors at some point outline 2 strategies for encoding atoms.
> A : The results for these strategies were included at the last page of the appendix! The results were found to be extremely similar.

---

### Author Response · Authors · 2022-08-02
**Changes**

We highlights hereafter some of the modifications that have been made to the paper :
We added a new experiment on the Perov-5 dataset. On this dataset the improvement with respect to the baselines is much more significant.
We added variance estimates for the evaluation metrics on the proposed models. These will also be added for the baseline models in the final version of the paper.
We improved the discussion of the results, mentioned some limits and expended on future direction
We added some discussion on why shy space groups are important and why equivariance and not only invariance should be considered.
We clarified some of the notation regarding groups
We expanded the details of the experiments and implementation in the appendix.

---

### Comment · Reviewer_MuHL · 2022-08-04
**Rebuttal acknowledged**

The authors have adequately addressed all concerns and shown great empirical results. I believe this is a strong paper with clean ideas and thorough execution. It's a method I'd be excited to use myself. I believe it fully deserves publication in this venue.

---

### Meta-Review · Area_Chair_FrWs · 2022-08-28

**Recommendation:** Accept
**Confidence:** Certain

**Metareview:**

This paper proposes an extension of recent work on equivariant graph neural networks to account for equivariance to crystalline symmetries. This work has the potential to be quite impactful since modeling crystalline structures is important and has received little attention compared to the modeling of molecular systems.

Two reviewers argued in favor of acceptance, citing the nontrivial nature of the problem and the solution. They also commented on the high quality of the writing. Lastly, the positive reviewers found the results promising after discussion. The negative reviewer focused on missing background material and lack of novelty of the proposed method. I am confident that the authors have / will continue to address the concerns of the negative referee by adding extra background materials and definitions for common terms. I believe that this paper makes enough novel progress on a difficult problem to be worth accepting to NeurIPS despite the fact that it builds on prior work.

**Award:**

No

---

### Decision · Program_Chairs · 2022-09-14

Accept